# Secondary structure of the SARS-CoV-2 genome is predictive of nucleotide substitution frequency

Zach Hensel*

ITQB NOVA, Universidade NOVA de Lisboa, Lisbon, Portugal

## eLife Assessment

This short article uses mutation counts in phylogenies of millions of SARS-CoV-2 genomes to show that mutation rates systematically differ between regions that are paired or unpaired in the predicted RNA secondary structure of the viral genome. Such an effect of pairing state is not unexpected, but its systematic demonstration using millions of viral genomes is **valuable** and **convincing**.

## Abstract
Accurate estimation of the effects of mutations on SARS-CoV-2 viral fitness can inform public-health responses such as vaccine development and predicting the impact of a new variant; it can also illuminate biological mechanisms including those underlying the emergence of variants of concern. Recently, Lan et al. reported a model of SARS-CoV-2 secondary structure and its underlying dimethyl sulfate reactivity data (Lan et al., 2022). I investigated whether base reactivities and secondary structure models derived from them can explain some variability in the frequency of observing different nucleotide substitutions across millions of patient sequences in the SARS-CoV-2 phylogenetic tree. Nucleotide basepairing was compared to the estimated 'mutational fitness' of substitutions, a measurement of the difference between a substitution's observed and expected frequency that is correlated with other estimates of viral fitness (Bloom and Neher, 2023). This comparison revealed that secondary structure is often predictive of substitution frequency, with significant decreases in substitution frequencies at basepaired positions. Focusing on the mutational fitness of C→U, the most common type of substitution, I describe C→U substitutions at basepaired positions that characterize major SARS-CoV-2 variants; such mutations may have a greater impact on fitness than appreciated when considering substitution frequency alone.

*For correspondence:
zach.hensel@itqb.unl.pt

Competing interest: The author declares that no competing interests exist.

## Introduction

While investigating the significance of the substitution C29095U, detected in a familial cluster of SARS-CoV-2 infections (*Chan et al., 2020*), I hypothesized that this synonymous substitution reflected the high frequency of C→U substitution during the pandemic (*De Maio et al., 2021*). Specifically, frequent C29095U substitution had previously complicated attempts to infer recombinant genomes (*VanInsberghe et al., 2021*). Preliminary investigation revealed that C29095U was the fourth most frequent C→U substitution, occurring almost seven times as often as a typical C→U substitution (*Bloom and Neher, 2023*). While there was no clear reason for selection of this synonymous substitution, C29095 was found to be unpaired in a secondary structure model (*Sun et al., 2021*). The 5' context of C29095 is UC, and additional SARS-CoV-2 secondary structure models that I consulted (see below) agree that C29095 occurs as a single unpaired base in an asymmetrical internal loop, opposite three unpaired bases. Although one study found that exogenous expression of APOBEC3A

**eLife digest** Tracking and characterizing mutations in a fast-evolving virus such as SARS-CoV-2, the cause of COVID-19, are valuable for an effective public health response and for developing treatments and vaccines that work well against mutated viruses. Mutations in the SARS-CoV-2 genome happen when errors are made as the RNA genetic material is copied.

The SARS-CoV-2 genome is made of an RNA chain consisting of four units called nucleotides, known as A, C, G, and U. Unlike DNA, where all nucleotides of one strand are paired with the other strand, RNA usually consists of a single strand. However, the SARS-CoV-2 genome folds into a compact and dynamic structure where some RNA nucleotides form pairs with others within the same strand, similar to the double-stranded DNA molecules.

Many mutations have no effect, but some can increase the ability of the virus to spread from person to person. This increased 'viral fitness' is difficult to predict, but understanding which mutations are more likely to occur can help scientists estimate risks. For example, a common type of mutation causes one RNA nucleotide – A, U, C or G – to be replaced by another. In SARS-CoV-2, the most common mutations involve C being replaced by U.

To find out if the pairing of RNA nucleotides affects how likely it is for a mutation to occur, Hensel looked at the rates of tens of thousands of different mutations, previously estimated from millions of publicly available sequences of the SARS-CoV-2 genome. He analyzed how often different types of mutations happen in paired versus unpaired RNA nucleotides. Only mutations that change the nucleotide sequence and not the protein sequences (so-called 'neutral' mutations) were considered in this analysis.

Hensel observed that C to U and G to U mutations are about four times more likely in unpaired nucleotides compared to paired ones. Other types of 'neutral' mutations, like A to G, were found to be equally likely, regardless of being paired or unpaired. Understanding that certain unpaired nucleotides mutate more often can help explain why some mutations are much more common than others. This should be considered in future models of the evolution of SARS-CoV-2 and other RNA viruses. A better understanding of mutation rates will also inform vaccine development and improve our estimates of viral fitness, potentially improving future public health responses.

accelerated C→U substitution in SARS-CoV-2-infected cells in the UC context (*Nakata et al., 2023*), the most frequently mutated positions were in the context of larger unpaired regions.

I hypothesized that non-enzymatic deamination may be more frequent for unpaired cytosine residues, which was supported by a previous analysis with a resolution of approximately 300 nucleotides (*Li et al., 2023*) and by a comparison of sequence diversity in early SARS-CoV-2 genomes to an RNA folding prediction (*Simmonds, 2020*). However, these studies differed in their interpretations of relatively high C→U substitution frequencies at unpaired positions, leaving it unresolved whether this was primarily driven by higher mutation rates at unpaired positions or fitness costs of mutations disrupting secondary structure. To determine whether secondary structure was in fact correlated with mutation frequency at single-nucleotide resolution and to extend this analysis beyond C→U substitutions, the dataset reported by Lan et al. was compared to the mutational fitness estimates reported by *Bloom and Neher, 2023*. Note that 'mutational fitness' is not a measurement of viral fitness per se; rather, it is an estimate made assuming that the expected frequencies of neutral mutations are determined only by the type of substitution (with C→U being much more frequent than all other types of substitutions).

## Results

There was a significant increase in synonymous substitution frequencies at unpaired positions for C→U, G→U, C→A, and U→C, but not for A→G or G→A ($p<0.05$; Tukey's range test with Bonferroni correction; A→U, G→C, and C→G were also significant in an unplanned analysis of all substitution types). For all substitution types with significant differences, unpaired substitution frequencies were higher than basepaired substitution frequencies. The largest effects were observed for C→U and G→U (*Figure 1*). In this secondary structure model, there is basepairing for 60% and 73% of C and G positions, respectively (limited to those covered in the mutational fitness analysis). The largest impacts on

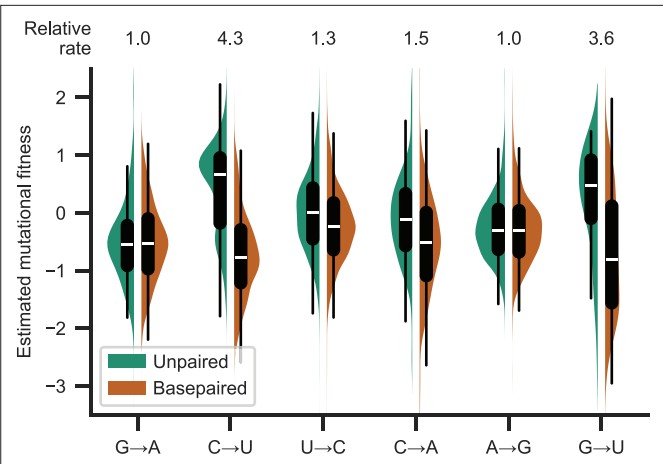

**Figure 1.** Basepairing is predictive of synonymous substitution frequency. Distribution of frequencies of synonymous substitutions for the most common substitutions (each approximately corresponding to 5% or more of observed substitutions), expressed as the estimated mutational fitness, which is a logarithmic comparison of the observed versus the expected number of occurrences of each type of substitution in the SARS-CoV-2 phylogenetic tree (*Bloom and Neher, 2023*). Distributions are grouped by substitution type and whether positions are basepaired in a full-genome secondary structure model of SARS-CoV-2 in Huh7 cells (*Lan et al., 2022*). Boxplots indicate the median and interquartile range. The median relative substitution rate (ratio of median rates of substitutions at unpaired and paired positions) is shown above each type of substitution.

The online version of this article includes the following figure supplement(s) for figure 1:

**Figure supplement 1.** Basepairing dependence of effect of 5′ context on synonymous C→U substitution frequency.

**Figure supplement 2.** Basepairing dependence of effect of 3′ context on synonymous C→U substitution frequency.

median estimated mutational fitness were observed for synonymous C→U and G→U mutations, with apparent fitness higher at unpaired positions than at paired positions by 1.44 and 1.25, respectively. Expressed in terms of substitution frequency rather than mutational fitness (*Figure 1*), the frequency of synonymous C→U and G→U substitutions was about four times higher at unpaired positions than at basepaired positions. Together, this demonstrates a meaningful impact of secondary structure on substitution frequencies.

Similar results were reported by two studies after this study was first preprinted. One study identified a basepairing dependence for C→U substitutions during serial passage of different SARS-CoV-2 variants and considered mutational fitness estimates from Bloom and Neher for more rare substitution types (*Gout et al., 2024*). In another analysis that analyzed recurrent substitutions in multiple lineages, little difference was observed overall for C→U substitutions in different 5′ and 3′ contexts (*Simmonds, 2024*). It was observed that only the few C→U substitutions that appeared with the highest frequencies had a strong preference for 5′ U context. *Figure 1—figure supplement 1* shows that the degree of this 5′ U preference becomes more clear when limiting analysis to synonymous substitutions and separately analyzing basepaired positions (which show no 5′ preference) and unpaired positions. At unpaired positions, synonymous C→U substitutions with 5′ U context have a higher apparent fitness (0.83 [0.79, 0.89], 95% CI from 1000 bootstrapped samples) than with 5′ G (0.15 [0.01, 0.50]), 5′A (0.55 [0.45, 0.68]), or 5′ C (0.57 [0.33, 0.69]). Conversely, synonymous C→U substitution rates with 3′ G context were elevated at both unpaired and paired positions (*Figure 1—figure supplement 2*), suggesting some CpG suppression with little dependence on basepairing. Notably, the previously mentioned C29095 is both unpaired and appears in the context UCG.

Basepairing in the secondary structure model appeared to be more predictive of estimated mutational fitness than average dimethyl sulfate (DMS) reactivity, with correlation coefficients of 0.59 [0.55, 0.62] and 0.45 [0.40, 0.49], respectively, for synonymous C→U substitutions (point biserial and Spearman correlation coefficients; 95% CIs from 1000 bootstrapped samples). However, DMS reactivity was more strongly correlated with estimated mutational fitness than basepairing when analysis

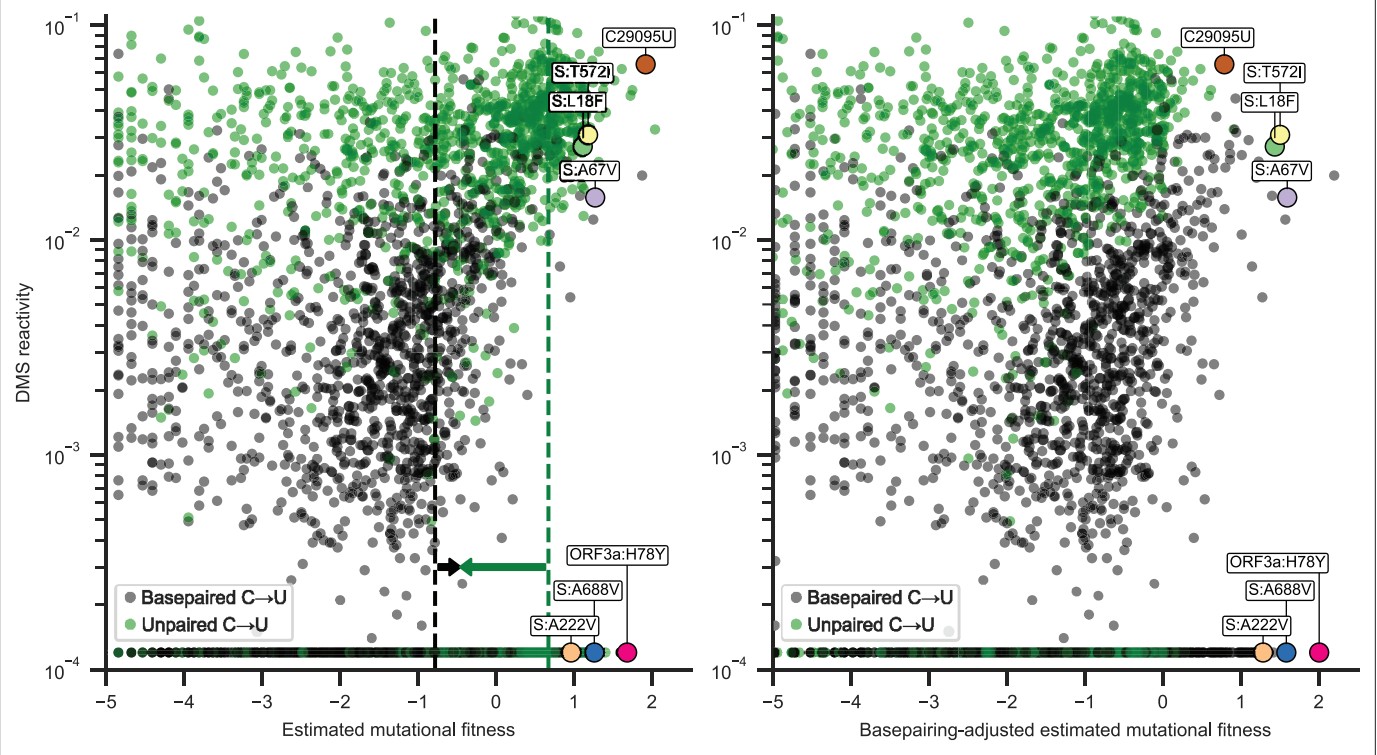

**Figure 2.** Estimated mutational fitness correlates with secondary structure for nonsynonymous C→U substitutions. Scatter plots compare mutational fitness to average dimethyl sulfate (DMS) reactivity for positions with potential nonsynonymous C→U substitutions. The minimum observed DMS reactivity value is assigned to positions lacking data. Points are colored by basepairing in the full genome secondary structure model. Nonsynonymous C→U substitutions at basepaired positions are highlighted, which rank highly for mutational fitness and characterize major SARS-CoV-2 lineages. Synonymous C29095U at an unpaired position is also highlighted. Left: estimated mutational fitness based only on observed versus expected occurrences of C→U at each position. Dashed lines indicate the median estimated mutational fitness for synonymous substitutions at paired and unpaired positions. Arrows indicate the magnitudes of adjustments made to mutational fitness that result in median fitness of synonymous substitutions at paired (+0.32) and unpaired (−1.13) positions identical to the unadjusted median for all synonymous substitutions (−0.46). Right: mutational fitness adjusted by constants derived from the medians of mutational fitness for synonymous substitutions at basepaired, unpaired, and all potential C→U positions.

The online version of this article includes the following figure supplement(s) for figure 2:

**Figure supplement 1.** Estimated mutational fitness correlates with secondary structure for synonymous C→U substitutions.

was limited to positions with detectable DMS reactivity (0.77 [0.74, 0.79], excluding the sites plotted at the minimum measured value of 0.00012). This suggests that considering structural heterogeneity captured in base reactivity datasets could improve predictions of substitution rates. Correlation coefficients remained significant, but were reduced (0.17 [0.14, 0.20] and 0.13 [0.09, 0.16]) when considering nonsynonymous C→U substitutions (*Figure 2*, left), consistent with larger and often negative effects of nonsynonymous mutations on viral fitness (*Bloom and Neher, 2023*).

Since basepairing is correlated with substitution rates, the degree of correlation between secondary structure models and substitution rates can be used to compare models that differ in underlying experimental methods and analysis. In light of this, secondary structure models derived from two DMS reactivity datasets (from SARS-CoV-2 isolate WA1 infection of Huh7 and Vero cells) (*Lan et al., 2022*) and two SHAPE reactivity datasets (SARS-CoV-2 isolates WA1 and Leiden0002 infection of Huh7.5 and Vero cells, respectively) (*Huston et al., 2021*; *Manfredonia et al., 2020*) were compared by degree of correlation with estimated mutational fitness. Correlations between basepairing and synonymous C→U substitution rates were highest for models derived from DMS data (0.59 [0.55, 0.62] and 0.63 [0.60, 0.67] for Huh7 and Vero datasets, respectively). Models derived from SHAPE data had modestly lower correlations (0.55 [0.50, 0.59] and 0.50 [0.46, 0.54] for Huh7 and Vero datasets, respectively). However, the methods used to construct each of these four models have difference

beyond base reactivity probes, cell type, and viral isolate, so small differences in correlation cannot be attributed to a single factor.

Thus, the high mutation rate of C29095U could be more accurately estimated by considering DMS reactivity than basepairing alone (*Figure 2*) since it is in the 99th percentile of DMS activity for positions with synonymous C→U substitutions. No major difference in this trend was observed across the SARS-CoV-2 genome. As a first-order approximation, two constants were calculated to equalize median mutational fitness for synonymous substitutions at basepaired, unpaired, and all positions. An 'adjusted mutational fitness' can then be calculated for C→U substitutions by incrementing mutational fitness by +0.32 at basepaired position and by -1.13 at unpaired positions (results were similar when considering fourfold degenerate positions rather than all synonymous substitutions). Scatter plots compare DMS reactivity to estimated mutational fitness at positions with nonsynonymous C→U substitutions before and after applying this coarse adjustment (*Figure 2*, left and right).

This adjustment equalizes median mutational fitness for synonymous C→U substitutions (*Figure 2— figure supplement 1*), and remaining correlation between 'adjusted mutational fitness' and DMS reactivity (0.18 [0.13, 0.22]) shows that this coarse adjustment could be improved with more complex analysis. However, a binary model of basepairing alone is sufficient to capture much of the variation in synonymous substitution rates and could be extended to other substitution types with rates depending on substitution types that lack reactivity data.

For a preliminary estimate of whether nonsynonymous C→U substitutions at basepaired positions are prone to underestimation of mutational fitness, I tested the hypothesis that C→U having highly ranked fitness at basepaired positions would be mutations that characterize significant SARS-CoV-2 lineages (arbitrarily defined as having 5% prevalence in the 1-week average of global sequences on the CoV-Spectrum website [*Chen et al., 2022*] at any time during the pandemic). This was the case for 6 of the top 15 C→U substitutions at basepaired positions; their encoded mutations are shown in *Figure 2*. Top-ranked mutations characterize Omicron lineages BA.1, one of the first recognized recombinant lineages XB, Gamma P.1, and lineage JN.1.7, which had over 20% global prevalence in spring 2024. Half of these mutations have relatively high DMS reactivity for basepaired positions and half have very low DMS reactivity. By comparison, the synonymous substitution C29095U at an unpaired position has very high estimated mutational fitness and DMS reactivity. Despite having a higher median estimated mutational fitness (1.41 vs 1.10), only 3 of the top 15 nonsynonymous C→U at unpaired positions define major lineages (BQ.1.1, JN.1.8.1, and BA.2.86.1).

Of particular note is C22227T at a basepaired position encoding the spike A222V mutation. This was one of the mutations that characterized the B.1.177 lineage. This lineage, also known as EU1, characterized a majority of sequences in Spain in summer 2020 and eventually in several other countries in Europe prior to the emergence of the Alpha variant. However, it was unclear whether this lineage had higher fitness than other lineages or if A222V specifically conferred a fitness advantage (*Hodcroft et al., 2021*). Further investigation as well as its recurrence in the major Delta sublineage AY.4.2 provided additional evidence of an increase in viral fitness and suggested molecular mechanisms (*Ginex et al., 2022*). Here, I focus on C→U substitutions for comparison to DMS reactivity data, but I also note that top-ranked G→U substitutions at basepaired positions are rich in mutations to ORF3a and also include mutations that characterize variants of concern, such as nucleocapsid D377Y in Delta. Lastly, note that, following the coarse adjustment for basepairing inferred from synonymous substitutions, nonsynonymous C→U substitutions characterizing major variants now have some of the highest estimated mutational fitness for C→U substitutions (*Figure 2*, right).

## Discussion

This analysis shows that it is informative to combine apparent viral fitness, inferred from millions of patient SARS-CoV-2 sequences published during the pandemic, with accurate secondary structure measurements. It is important to remember that apparent 'mutational fitness' results from a combination of the rates at which diversity is generated and the subsequent selection processes. Importantly, genome secondary structure can impact both. However, even the unprecedented density of sampling SARS-CoV-2 genomes has been insufficient to reliably infer fitness impacts of single mutations more directly from dynamics subsequent to occurrences in the SARS-CoV-2 phylogenetic tree (*Bloom and Neher, 2023*; *Richard et al., 2021*).

Further investigation into phenomena reported here, such as the lack of apparent secondary structure dependence for A→G and G→A substitutions, could inform investigation of underlying mutation and selection mechanisms. An analysis focused on basepairing dependence of C→U substitution frequencies considered that this may be driven by conservation of secondary structure (*Gout et al., 2024*). However, the lack of basepairing dependence for A→G and G→A substitutions supports the hypothesis that this effect, on average, arises from differences in chemical reaction rates (*Li et al., 2023*). Nonetheless, future analyses could consider whether substitution rates have different basepairing dependencies in regions in which synonymous mutations exhibit fitness reductions (*Bloom and Neher, 2023*). The extents of enzymatic and non-enzymatic mutation processes driving differences in substitution rates remains unclear (*Simmonds, 2024*). Comparing the degree to which rates depend on secondary structure could provide insight to support different mutation mechanisms. A recent in vitro study suggested that oxidation of guanine can give rise to G→U substitutions (*Akagawa et al., 2024*), which, like C→U substitutions, are both very common in human SARS-CoV-2 infections and exhibit strong basepairing dependence.

I suggest that secondary structure, along with other data correlating with substitution frequency, can be used to refine estimates of mutational fitness both for future SARS-CoV-2 variants and for other viruses. More sophisticated analysis can incorporate structural heterogeneity (*Lan et al., 2022*) as well as local sequence context (*Lamb et al., 2024*). *Figure 2—figure supplement 1* shows how the coarse adjustment described in this article does not capture all information available in DMS reactivity data, and DMS interrogates adenine and cytosine basepairing while SHAPE reagents can be used to investigate backbone flexibility for all nucleotides. Clearly, datasets from different experiments using different chemical probes can be combined to produce more accurate estimates of apparent mutational fitness. Furthermore, additional measurements of secondary structure for genomes of new variants or modeling may reveal significant changes to secondary structure since 2020. This could be evident in in vitro substitution rates for different SARS-CoV-2 variants (*Gout et al., 2024*). Lastly, secondary structures determined from in vitro chemical probing experiments may differ from secondary structures in vivo. For the spike protein, the correlation between estimated mutational fitness and pseudovirus entry quantified by deep mutational scanning serves as one metric that can be used to optimize models (*Lamb et al., 2024*). However, it is critical to evaluate uncertainty in any model estimating fitness of a new variant. Initial estimates can be refined by rapid in vitro characterization and continued genomic surveillance, helping shed light on which mutations contribute to viral fitness for potential variants of concern (*Carabelli et al., 2023*).

## Materials and methods

The data compared in this study consisted of estimated mutational fitness for the SARS-CoV-2 genome as reported by *Bloom and Neher, 2023* (supplementary data: `ntmut_fitness_all.csv`; November 2024 dataset) as well as population-averaged DMS reactivities for SARS-CoV-2-infected Huh7 cells and the corresponding secondary structure model reported by *Lan et al., 2022* (supplementary data 7 and 8). Two other secondary structure models based upon SHAPE reactivity data were also used (*Huston et al., 2021*; *Manfredonia et al., 2020*). Note that the estimated mutational fitness is logarithmically related to the ratio of the observed and expected number of occurrences of a nucleotide substitution, with large and asymmetric differences in the frequencies of different types of synonymous substitutions (*De Maio et al., 2021*). Additionally, note that DMS data was obtained in experiments using the WA1 strain in Lineage A, which differs from the more common Lineage B at three positions and could have different secondary structure. Furthermore, mutational fitness is estimated from the phylogenetic tree of published sequences (the public UShER tree [*Turakhia et al., 2021*] additionally curated to filter likely artifacts such as branches with numerous reversions) that are typically far more divergent and subsequently will have somewhat different secondary structures. Since the dataset used for mutational fitness aggregates data across viral clades, my analysis will not capture secondary structure variation between clades or indels and masked sites that were not considered in that analysis (*Bloom and Neher, 2023*). I focused on the most common types of nucleotide substitutions: those comprising approximately 5% or more of total substitutions.

## Acknowledgements

Although I did not directly access any genome sequencing databases for this work, I am grateful to the patients who volunteered samples, and to the clinicians, technicians, and teams behind the databases who have made sequencing data available so that this work is possible. I thank Erol Akcay, Alexander Crits-Christoph, Florence Débarre, Ryan Hisner, and James Yates for critical comments on the manuscript and discussions. ZH is supported by Fundação para a Ciência e a Tecnologia (FCT) through MOSTMICRO-ITQB (DOI 10.54499/UIDB/04612/2020; DOI 10.54499/UIDP/04612/2020) and LS4FUTURE Associated Laboratory (DOI 10.54499/LA/P/0087/2020).

## Additional information

### Funding

| Funder | Grant reference number | Author |
|---|---|---|
| Fundação para a Ciência e a Tecnologia | 10.54499/UIDB/04612/2020 | Zach Hensel |
| Fundação para a Ciência e a Tecnologia | 10.54499/UIDP/04612/2020 | Zach Hensel |
| Fundação para a Ciência e a Tecnologia | 10.54499/LA/P/0087/2020 | Zach Hensel |

The funders had no role in study design, data collection and interpretation, or the decision to submit the work for publication.

### Author contributions

Zach Hensel, Investigation, Visualization, Writing – original draft

### Author ORCIDs

Zach Hensel ⓘ https://orcid.org/0000-0002-4348-6229

Reviewer #1 (Public review): https://doi.org/10.7554/eLife.98102.3.sa1
Reviewer #2 (Public review): https://doi.org/10.7554/eLife.98102.3.sa2
Author response https://doi.org/10.7554/eLife.98102.3.sa3

## Additional files

### Supplementary files

MDAR checklist

### Data availability

Data analyzed in this manuscript and a Python notebook to reproduce analysis and figures in this manuscript are available at https://github.com/smmlab/SARS2-fitness-secondary-structure (copy archived at *smmlab, 2024*). The underlying data was obtained from supplementary files of *Bloom and Neher, 2023* and *Lan et al., 2022* as described in Methods section.

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
