## [Editor Report · eLife Assessment]

This short article uses mutation counts in phylogenies of millions of SARS-CoV-2 genomes to show that mutation rates systematically differ between regions that are paired or unpaired in the predicted RNA secondary structure of the viral genome. Such an effect of pairing state is not unexpected, but its systematic demonstration using millions of viral genomes is **valuable** and **convincing**.

---

## [Referee Report · Reviewer #1 (Public review)]

Summary:

This very short paper shows a greater likelihood of C->U substitutions at sites predicted to be unpaired in the SARS-CoV-2 RNA genome, using previously published observational data on mutation frequencies in SARS-CoV-2 (Bloom and Neher, 2023).

General comments:

A preference for unpaired bases as target for APOBEC-induced mutations has been demonstrated previously in functional studies so the finding is not entirely surprising. This of course assumes that A3A or other APOBEC is actually the cause of the majority of C->U changes observed in SARS-CoV-2 sequences.

I'm not sure why the authors did not use the published mutation frequency data to investigate other potential influences on editing frequencies, such as 5' and 3' base contexts. The analysis did not contribute any insights into the potential mechanisms underlying the greater frequency of C->U (or G->U) substitutions in the SARS-CoV-2 genome.

Comments on revisions:

The revisions have addressed my main comments in my review.

---

## [Referee Report · Reviewer #2 (Public review)]

Hensel investigated the implications of SARS-CoV-2 RNA secondary structure in synonymous and nonsynonymous mutation frequency. The analysis integrated estimates of mutational fitness generated by Bloom and Neher (from publicly available patient sequences) and a population-averaged model of RNA base-pairing from Lan et al (from DMS mutational profiling with sequencing, DMS-MaPseq)

The results show that base-pairing limits the frequency of some synonymous substitutions (including the most common C→T), but not all: G→A and A→G substitutions seem unaffected by base-pairing.

The author then addressed nonsynonymous C→T substitutions at basepaired positions. While there is still a generally higher estimated mutational fitness at unpaired positions, they propose a coarse adjustment to disentangle base-pairing from inherent mutational fitness at a given position. This adjustment reveals that nonsynonymous substitutions at base-paired positions, which define major variants, have higher mutational fitness.

Overall, this manuscript highlights the importance of considering RNA secondary structure in viral evolution studies.

The conclusions of this work are generally well supported by the data presented. Particularly, the author acknowledges most limitations of the analyses and addresses them. Even though no new sequencing results were generated, the author used available data generated from the analysis of roughly seven million sequenced patient samples. Finally, the author discusses ways to improve the current available models.

There are a number of limitations of this work that should be highlighted, specifically in regard to the secondary structure data used in this paper. The Lan et al. dataset was generated using a multiplicity of infection (MOI) of 0.05, 24 hours post-infection (h.p.i.). At such a low MOI and late timepoint, viral replication is not synchronous and sequencing artifacts might be generated by cell debris and viral RNA degradation, therefore impacting the population-averaged results. In addition, the nonsynonymous base-paired positions in Figure 2 have relatively high population-averaged DMS reactivity, which suggests those positions are dynamic. Therefore, the proposed adjustment could result in an incorrect estimation of their inherent mutational fitness.

Additionally, like all such RNA probing experiments within cells, it remains difficult to deconvolve DMS/SHAPE low reactivity with RNA accessibility (e.g. from protein binding).

This work presents clear methods and an easy-to-access bioinformatic pipeline, which can be applied to other RNA viruses. Of note, it can be readily implemented in existing datasets. Finally, this study raises novel mechanistic questions on how mutational fitness is not correlated to secondary structure in the same way for every substitution.

Overall, this work highlights the importance of studying mutational fitness beyond an immune evasion perspective. On the other hand, it also adds to the viral intrinsic constraints to immune evasion.

Comments on revisions:

Following revision by the author, our concerns have been addressed. The additional analysis strengthens the conclusions & the revisions to the text have improved the manuscript for a general audience.

---

## [Author Response]

The following is the authors’ response to the original reviews.

**Reviewer 1 Public Review:**
SummaryThis very short paper shows a greater likelihood of C->U substitutions at sites predicted to be unpaired in the SARS-CoV-2 RNA genome, using previously published observational data on mutation frequencies in SARS-CoV-2 (Bloom and Neher, 2023).General commentsA preference for unpaired bases as a target for APOBEC-induced mutations has been demonstrated previously in functional studies so the finding is not entirely surprising. This of course assumes that A3A or other APOBEC is actually the cause of the majority of C->U changes observed in SARS-CoV-2 sequences.I'm not sure why the authors did not use the published mutation frequency data to investigate other potential influences on editing frequencies, such as 5' and 3' base contexts. The analysis did not contribute any insights into the potential mechanisms underlying the greater frequency of C->U (or G->U) substitutions in the SARS-CoV-2 genome.

I have added additional discussions of mechanisms focusing on the question of whether basepairing bias is primarily driven by secondary structure dependence of underlying mutation rates or by conservation of secondary structure (Discussion lines 178–192) and I added a brief analysis of the 5′ and 3′ contexts of the relationship between being basepaired in a secondary structure model and apparent mutational fitness (Figures S1 and S2, Results lines 85–97). I found that the 5′ context of unpaired, but not paired basepairs influences apparent mutational fitness (preference for 5′ U), and that the is also . Additionally, there is a 3′ preference for G, indicating some CpG suppression. This contrasts to some degree with another analysis based on counting lineage frequencies that may have lacked power to detect relatively small effects (Simmonds *mBio* 2024).

**Reviewer 1 Author recommendations:**
There are at least 5 publications describing the mapping/prediction of SARS-CoV-2 RNA secondary structure from 2022-2023 and their predictions are not entirely consistent. Why did the authors only refer to the Lan et al. paper?

I have added comparisons when the Lan et al secondary structure model is replaced by one of two others derived from SHAPE data (Results lines 110–122). Unsurprisingly, similar secondary structure models give similar results and performance is modestly higher for the models from Lan *et al*. This is consistent with their observations that DMS reactivities performed better as classifiers of SL5 and ORF1 secondary structure (the reason I compared to this secondary structure model and reactivity data set rather than others), but I did not go into detail on this in the revision since there are many differences in methods beyond class of reactivity probe. For example, somewhat stronger correlation for the Vero than the Huh7 dataset in Lan *et al* could arise from combining data from two replicates, from cell type, or from differences in data analysis methods. It’s also a small difference and cannot be confidently distinguished from noise.

I conducted a preliminary comparison of the performance of DMS and SHAPE data for predicting mutations where DMS data is available, but I opted against including this analysis in the manuscript for the same reasons. Instead, I included in results and discussion comments on how, in general, reactivity data contains information that is predictive of substitution rates that is not captured by binary secondary structure models. I also discuss how multiple data sources can potentially be integrated to more accurately predict the impact of a substitution on fitness (Discussion lines 195–201).

Specific substitutions are referred to as C->T and C29085T for example, but as the genome of SARS-CoV-2 is RNA, and T should be a U.

I agree and I have changed all “T” to “U” in the paper and analysis scripts. The choice of “T” was motivated by what seemed to appear most frequently in papers on SARS-CoV-2 mutational spectra, but “U” is nearly universal in papers on secondary structure and mutation mechanisms, so I agree it makes more sense in this paper.

The C29085T substitution is somewhat non-canonical as it is a single base bulge in a longer duplex section of dsRNA, very unlike the favoured sites for mutation in the Nakata et al paper.

I have added a discussion of Nakata *et al* (NAR 2023) (Introduction lines 29–32). I did not go into this depth in the revision, but the analysis of ~2M patient sequences in Nakata *et al* also noted a high rate of UUC→UUU substitution, so the UUUC context of C29095 (shared by 3 of the 10 positions highlighted in Nakata *et al* that had high mutation frequencies with exogenous APOBEC3A expression) could be interesting to investigate further.

High C29095U substitution frequency is indeed somewhat at odds with the results in that work, which found that UC→UU substitutions to be elevated in longer single-stranded regions than the context of C29095U in SARS-CoV-2 secondary structure models (a single unpaired base opposing three unpaired bases in an asymmetric internal loop).

I'm not sure why DMS reactivity is considered a separate variable from pairing likelihood as one informs the other.

The intent here, which was not clear, was to show that a binary basepairing model that uses DMS reactivities as constraints does not capture all of the information available. I have clarified this in as described above discussing information in different reactivy datasets.

The C29095U substitution is also relavent to the consideration of DMS reactivity in addition to the resulting secondary structure model. These are not considered as separate predictors and the reason for showing both is mentioned in the paper: “DMS reactivity was more strongly correlated with estimated mutational fitness than basepairing when analysis was limited to positions with detectable DMS reactivity.” I have clarified this in the revised manuscript and also it is relevant to the discussion of a potential model integrating all available datasets.

**Reviewer 2 Public Review:**
Hensel investigated the implications of SARS-CoV-2 RNA secondary structure in synonymous and nonsynonymous mutation frequency. The analysis integrated estimates of mutational fitness generated by Bloom and Neher (from publicly available patient sequences) and a population-averaged model of RNA basepairing from Lan et al (from DMS mutational profiling with sequencing, DMS-MaPseq).The results show that base-pairing limits the frequency of some synonymous substitutions (including the most common CT), but not all: GA and AG substitutions seem unaffected by base-pairing.The author then addressed nonsynonymous CT substitutions at base-paired positions. While there is still a generally higher estimated mutational fitness at unpaired positions, they propose a coarse adjustment to disentangle base-pairing from inherent mutational fitness at a given position. This adjustment reveals that nonsynonymous substitutions at base-paired positions, which define major variants, have higher mutational fitness.Overall, this manuscript highlights the importance of considering RNA secondary structure in viral evolution studies.The conclusions of this work are generally well supported by the data presented. Particularly, the author acknowledges most limitations of the analyses, and addresses them. Even though no new sequencing results were generated, the author used available data generated from the analysis of roughly seven million sequenced patient samples. Finally, the author discusses ways to improve the current available models.There are a number of limitations of this work that should be highlighted, specifically in regard to the secondary structure data used in this paper. The Lan et al. dataset was generated using a multiplicity of infection (MOI) of 0.05, 24 hours post-infection (h.p.i.). At such a low MOI and late timepoint, viral replication is not synchronous and sequencing artifacts might be generated by cell debris and viral RNA degradation, therefore impacting the population-averaged results. In addition, the nonsynonymous base-paired positions in Figure 2 have relatively high population-averaged DMS reactivity, which suggests those positions are dynamic. Therefore, the proposed adjustment could result in an incorrect estimation of their inherent mutational fitness.

I would go further than this to say that the proposed adjustmentment *will usually* result in an incorrect estimate. My intent is to propose an improved, but still likely incorrect, estimate by utilizing in vitro data to refine baseline mutation rates in order to obtain improved, but only coarsely adjusted, estimates of mutational fitness. I added a note in the discussion that in vitro reactivities (and, consequently, secondary structure models) may not reflect secondary structures in vivo (Discussion lines 204–205). I did not go into detail regarding the specific technical considerations mentioned here because they are outside the scope of my expertise.

I am not sure that top-ranked non-synonymous C→U positions have particularly high DMS values after coarse adjustment for basepairing (labeled amino acid mutations in Figure 2). Of the six common mutations used as examples, three have minimum values in the dataset considered (which is processed normalized/filtered data rather than raw data) and three do not have very high DMS reactivity.

However, there is clearly information in base reactivity that is not captured by a binary basepairing model, which is indicated by residual positive correlation between DMS reactivity and mutational fitness after adjustment. I now include a figure demonstrating this for synonymous C→U substitutions as Figure S3, and I have tried to clarify the language throughout the manuscript to make it clear that a more accurate adjustment is possible.

Additionally, like all such RNA probing experiments within cells, it remains difficult to deconvolve DMS/SHAPE low reactivity with RNA accessibility (e.g. from protein binding).

I agree, and in revising this manuscript it was interesting to see that Nakata *et al* (discussed above) identified relatively large single-stranded regions with enhanced UC→UU substitution frequencies with exogenous APOBEC3A expression, while C29095U, for example, is a single unpaired base with high DMS reactivity and high empirical C→U substitution frequency (discussed briefly in the introduction of the revised manuscript). Future analyses could consider heterogeneity in secondary structure as well as secondary structures with low heterogeneity where strained conformations could have higher reactivity.

This work presents clear methods and an easy-to-access bioinformatic pipeline, which can be applied to other RNA viruses. Of note, it can be readily implemented in existing datasets. Finally, this study raises novel mechanistic questions on how mutational fitness is not correlated to secondary structure in the same way for every substitution.Overall, this work highlights the importance of studying mutational fitness beyond an immune evasion perspective. On the other hand, it also adds to the viral intrinsic constraints to immune evasion.
**Reviewer 2 Author recommendations:**
Even though the experiment was not performed in this manuscript, it would be helpful for the readers if it was briefly explained how secondary structure is inferred from DMS reactivity, as this technique is not broadly used.It is not objective to refer to the Lan et al. model of RNA structure as "high quality" given the limitations of their experimental approach (low MOI, asynchronous infection, DMS-only, no long-range interactions) and the lack of external validation of the structure of the genome they propose.

I removed “high-quality” from the abstract. Since a result of the paper is that secondary structure correlates with synonymous substitution rates, this is an observation that can be used to retrospectively compare the quality of secondary structure models in this respect. I updated the manuscript to include such a comparison, and did not find a large difference between secondary structure models (Results lines 110–122). I added a discussion of how multiple data sources can potentially be integrated to more accurately predict the impact of a substitution of viral fitness.

I have also added a brief discussion of constraints on how much we can confidently infer from these experiments given limitations of the experimental approach. I note that DMS and SHAPE data provide information that can be combined to make a stronger model, and that predictions can be rapidly tested given observations by Gout (Symonds?) et al that in vitro substitution rates correlate with those observed during the pandemic (Discussion lines 195–201).

Mutational fitness from Bloom & Neher was derived throughout the pandemic, much of which came from a period with the most active surveillance (Delta / Omicron waves). Consequently, these viruses differ from the WA1 strain used by Lan et al. far more than the 3 nt differences between lineage A and B that the author refers to. The following sentence should therefore be revised to avoid misleading the reader:"Additionally, note that DMS data was obtained in experiments using the WA1 strain in Lineage A, which differs from the more common Lineage B at 3 positions and could have different secondary structure."

Revised:

“Additionally, note that DMS data was obtained in experiments using the WA1 strain in Lineage A, which differs from the more common Lineage B at 3 positions and could have different secondary structure. Furthermore, mutational fitness is estimated from the phylogenetic tree of published sequences (the public UShER tree (Turakhia et al., 2021) additionally curated to filter likely artifacts such as branches with numerous reversions) that are typically far more divergent and subsequently will have somewhat different secondary structures. Since the dataset used for mutational fitness aggregates data across viral clades, my analysis will not capture secondary structure variation between clades or indels and masked sites that were not considered in that analysis (Bloom and Neher, 2023).”

To determine the extent to which the results depend on the single RNA structure model, it would be informative "turn the crank again" on the analysis with one of the other RNA structure datasets for SARS-CoV-2 (though most other datasets suffer from similar problems of asynchronicity of infection).

I have added comparisons when the Lan *et al* secondary structure model is replaced by one of two others derived from SHAPE data as described above. Also, I conducted preliminary comparisons of underlying DMS and SHAPE reactivity data as described above, but I opted not to include these in the revised manuscript given that methods different beyond the chemical probe used. I also discuss how multiple data sources can potentially be integrated to more accurately predict the impact of a substitution of viral fitness.

In Figure 1 it would be helpful to add the values of the unpaired/basepaired ratios in the plot for clarity.Furthermore, a similar analysis using the substitution frequency, which strengthens the conclusions, is mentioned in the text, however, it is not shown. It could be shown as part of Figure 1, or as a supplementary figure.

This was a good suggestion since numbers around 1 are not perceived as being very significant. I added the ratio of median unpaired:paired rates to Figure 1, updated the corresponding manuscript text and the figure caption, and note that the numbers are somewhat changed from the first version of my manuscript because of updating to use the most up-to-date mutational fitness estimates.

It is not clear how the two constants were calculated to obtain the "adjusted mutational fitness". It could be shown as part of Figure 2, or as a supplementary figure.

I added dashed lines and arrows to Figure 2 showing median paired/unpaired mutational fitnesses and the adjustment made to normalize to the overall median. I also added Figure S3 showing this for synonymous substitutions, where it is more clear given the lower fraction of mutations with substantial fitness impacts.

Minor commentsStatements like "the current fast-growing lineage JN.1.7" never age well... please revise to state the period of time to which this refers.

Revised:

“…lineage JN.1.7, which had over 20% global prevalence in Spring 2024…”

Also, I checked the list of mutations and the examples given remain in the top 15 ranked basepaired, non-synonymous C→U mutations (BA.2-defining C26060U is added to the list, but I did not update to include this). It replaces C9246U, which was not mentioned in the first version of the manuscript.

Similarly, please provide context for the reader in the phrase: "This was one mutation that characterized the B.1.177 lineage" (e.g. add its early reference as "EU1" and that it predominated in Europe in autumn 2020, prior to the emergence of the Alpha variant).

Revised to add detail:

This was one of the mutations that characterized the B.1.177 lineage. This lineage, also known as EU1, characterized a majority of sequences in Spain in summer 2020 and eventually in several other countries in Europe prior to the emergence of the Alpha variant. However, it was unclear whether or this lineage had higher fitness than other lineages or if A222V specifically conferred a fitness advantage.

"massive sequencing of SARS-CoV-2" - the meaning of the word "massive" is unclear. Revise.

Revised “…millions of patient SARS-CoV-2 sequences published during the pandemic…”